# DEEP TEMPORAL GRAPH CLUSTERING

**Meng Liu**[1]    **Yue Liu**[1]    **Ke Liang**[1]    **Wenxuan Tu**[1]
**Siwei Wang**[2]    **Sihang Zhou**[1]    **Xinwang Liu**[1*]

[1]National University of Defense Technology, Changsha, China
[2]Intelligent Game and Decision Lab, Beijing, China
`mengliuedu@163.com, xinwangliu@nudt.edu.cn`

## ABSTRACT

Deep graph clustering has recently received significant attention due to its ability to enhance the representation learning capabilities of models in unsupervised scenarios. Nevertheless, deep clustering for temporal graphs, which could capture crucial dynamic interaction information, has not been fully explored. It means that in many clustering-oriented real-world scenarios, temporal graphs can only be processed as static graphs. This not only causes the loss of dynamic information but also triggers huge computational consumption. To solve the problem, we propose a general framework for deep Temporal Graph Clustering called TGC, which introduces deep clustering techniques to suit the interaction sequence-based batch-processing pattern of temporal graphs. In addition, we discuss differences between temporal graph clustering and static graph clustering from several levels. To verify the superiority of the proposed framework TGC, we conduct extensive experiments. The experimental results show that temporal graph clustering enables more flexibility in finding a balance between time and space requirements, and our framework can effectively improve the performance of existing temporal graph learning methods. The code is released: `https://github.com/MGitHubL/Deep-Temporal-Graph-Clustering`.

## 1   INTRODUCTION

Graph clustering is an important part of the clustering task, which refers to clustering nodes on graphs. Graph (also called network) data is common in the real world (Cui et al., 2018; Liang et al., 2023a; Hamilton, 2020), such as citation graphs, knowledge graphs, e-commerce graphs, etc. In these graphs, graph clustering techniques can be used for many applications, such as game community discovery, financial anomaly detection, urban criminal prediction, social group analysis, etc.

In recent years, deep graph clustering has received significant attention due to its ability to enhance the representation learning capabilities of models in unsupervised scenarios. Nevertheless, existing deep graph clustering methods mainly focus on static graphs and neglect temporal graphs. Static graph clustering methods treat the graph as the fixed data structure without considering the dynamic changes in graph. This means that in many clustering-oriented real-world scenarios, temporal graph data can only be processed as static graphs. But in these scenarios, there are usually a lot of dynamic events, where relationships and identities of nodes are constantly changing. Thus the neglect of time may lead to the loss of useful dynamic information.

Compared to static graphs, temporal graphs enable more fine observation of node dynamic interactions. However, existing temporal graph learning methods usually focus on link prediction rather than node clustering. This is because adjacency matrix-based deep graph clustering modules are no longer applicable to the batch-processing pattern based on the interaction sequence in temporal graphs. Thus we ask: what makes node clustering different between temporal graphs and static graphs? We attempt to answer this from the following perspectives.

---

*Corresponding Author: Xinwang Liu.

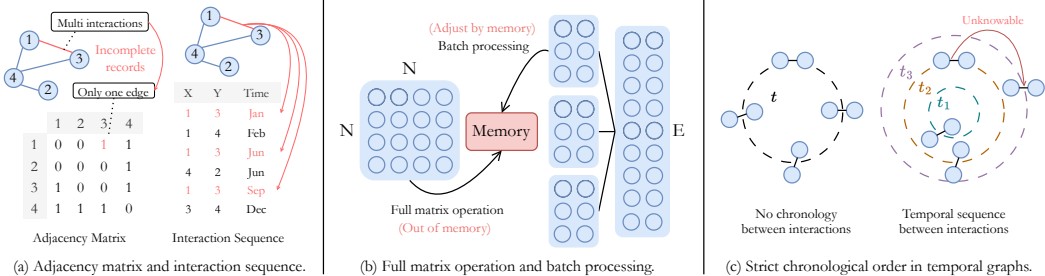

Figure 1: Difference between static graph and temporal graph.

(1) As shown in Fig. 1 (a), if there are several interactions between two nodes, the adjacency matrix can hardly reflect the dynamic changes, especially when these interactions belong to different timestamps. In contrast, temporal graphs utilize the interaction sequence (i.e., adjacency list) to store interactions. The existence of multiple interactions between nodes can be clearly represented by interaction sequence of the temporal graph, but it will be compressed into two forms of 0 or 1 in the adjacency matrix, i.e., it results in the absence of edges. This problem brings about a serious lack of information when there are frequent interactions between pairs of nodes.

(2) Further, we can find in Fig. 1 (b), static graph methods usually train the whole adjacency matrix, which is not convenient enough for large-scale graphs (i.e., possible out-of-memory problem). By slicing the interaction sequence into multiple batches, temporal methods are naturally suitable for large-scale data processing (i.e., batch size can adjust by memory). *And it also means that temporal methods can no longer take advantage of the node clustering modules based on the adjacency matrix.*

(3) Although there are a few static methods that split graph data into multiple sub-graphs to solve the out-of-memory problems, this still differs from temporal methods. The most important issue is that the loading of temporal graphs must strictly follow the chronological order, i.e., the earlier nodes cannot "see" the later nodes, as shown in Fig. 1 (c). Suppose there is an interaction between two nodes in a subgraph occurs in 2023, while an interaction between a subgraph and another subgraph occurs in 2020. So how do we reverse time the information from 2023 and pass it on in 2020? Therefore, we believe that our framework is a more appropriate solution in the field of temporal graphs. In this case, the temporal relationship of node interactions should still be taken into account during training.

Due to these discrepancies, the difficulty of temporal graph clustering is to find the balance between interaction sequence-based batch-processing patterns and adjacency matrix-based node clustering modules. Nowadays, few works discuss it comprehensively. Although a few methods refer to temporal graph clustering, they are incomplete and we discuss them in the Appendix.

Driven by this, we propose a general framework for Temporal Graph Clustering, called TGC. Such framework proposes two deep clustering modules (node assignment distribution and graph reconstruction) to suit the batch-processing pattern of temporal graphs. In addition, we discuss temporal graph clustering at several levels, including intuition, complexity, data, and experiment. To verify the superiority of the proposed framework TGC on unsupervised temporal graph representation learning, we conduct extensive experiments. The experimental results show that temporal graph clustering enables more flexibility in finding a balance between time and space requirements, and our framework can effectively improve the performance of existing temporal graph learning methods. In summary, our contributions are several-fold:

**Problem**. We discuss the differences between temporal graph clustering and static graph clustering.To the best of our knowledge, our work is the first to comprehensively focus on deep temporal graph clustering.

**Algorithm**. A simple general framework TGC is proposed, which introduces two deep clustering modules to suit the interaction sequence-based batch-processing pattern of temporal graphs.

**Dataset**. We discuss another issue that hinders the development of temporal graph clustering, namely the lack of datasets, and collate or develop several effective datasets.

**Evaluation**. We conduct several experiments to validate the clustering performance, flexibility, and transferability of TGC, and further elucidate the characteristics of temporal graph clustering.

## 2 METHOD

In this section, we first give the definitions of temporal graph clustering, and then describe the proposed framework TGC. Our framework contains two modules: a temporal module for time information mining, and a clustering module for node clustering. Here we introduce the classical HTNE (Zuo et al., 2018) method as the baseline of temporal loss, and further discuss the transferability of TGC on other methods in the experiments. For the clustering loss, we improve two node clustering technologies to fit temporal graphs, i.e., node-level distribution and batch-level reconstruction.

### 2.1 PROBLEM DEFINITION

If a graph contains timestamp records between node interactions, we call it a temporal graph.

**Definition 1.** *Temporal graph. Given a temporal graph $G = (V, E, T)$, where $V$ denotes nodes and $E$ denotes interactions. In temporal graphs, the concept of edges is replaced by interactions. Because although there is only one edge between two nodes, there may be multiple interactions that occur at different times. Multiple interactions between two nodes can be formulated as: $T_{x,y} = \{(x, y, t_1), (x, y, t_2), \cdots, (x, y, t_n)\}$. If two nodes interact with each other, we call them neighbors. When a node's neighbors are sorted by interaction time, its historical neighbor sequence can be formulated as: $N_x = \{(y_1, t_1), (y_2, t_2), \cdots, (y_l, t_l)\}$.*

Temporal graph clustering aims to group nodes into different clusters on temporal graphs.

**Definition 2.** *Temporal graph clustering. Node clustering in the graph follows some rules: (1) nodes in a cluster are densely connected, and (2) nodes in different clusters are sparsely connected. Here we define $K$ clusters to divide all nodes, i.e., $C = \{c_1, c_2, ..., c_K\}$. Node embeddings are continuously optimized during training and then fed into the K-means algorithm for performance evaluation during testing.*

### 2.2 BASELINE TEMPORAL LOSS

How to capture the dynamic information is the most important problem in temporal graphs. As a classical method, HTNE (Zuo et al., 2018) introduces the Hawkes process (Hawkes, 1971) to capture the dynamic information in graph evolution. Such a process argues that node future interaction are influenced by historical interactions and this influence decays over time. Given two nodes $x$ and $y$ interact at time $t$, their conditional interaction intensity $\lambda_{(x,y,t)}$ can be formulated as follows.

$$\lambda_{(x,y,t)} = \mu_{(x,y,t)} + h_{(x,y,t)} \tag{1}$$

According to Eq. 1, the conditional intensity can be divided into two parts: (1) the base intensity between two nodes without any external influences, i.e., $\mu_{(x,y,t)} = -||\boldsymbol{z}_x^t - \boldsymbol{z}_y^t||^2$, where $\boldsymbol{z}_x^t$ is the node embedding of $x$ at time $t$, and (2) the hawkes intensity $h_{(x,y,t)}$ from historical interaction influences, which is weighted by node similarity in addition to decaying over time.

$$h_{(x,y,t)} = \sum_{i \in N_x} \alpha_{(i,y,t)} \cdot \mu_{(i,y,t)}, \quad \alpha_{(i,y,t)} = \omega_{(i,x)} \cdot f(t_c - t_i) \tag{2}$$

Here, $\omega_{(i,x)}$ is the node similarity weight to evaluate a neighbor's importance in all neighbors. $f(t_c - t_i)$ denotes the influences from neighbors decay with time, and the earlier the time, the less the influence. $\delta_t$ is a learnable parameter, $t_c$ denotes the current timestamp.

$$\omega_{(i,x)} = \frac{\exp(\mu_{(i,x)})}{\sum_{i' \in N_x} \exp(\mu_{(i',x)})}, \quad f(t_c - t_i) = \exp(-\delta_t(t_c - t_i)) \tag{3}$$

Finally, given two nodes $x$ and $y$ interact at time $t$, their conditional intensity should be as large as possible, and the intensity of node $x$ with any other node should be as small as possible. Since it requires a large amount of computation to calculate the intensities of all nodes, we introduce the negative sampling technology (Mikolov et al., 2013), which samples several unrelated nodes as negative samples. Thus the baseline temporal loss function can be calculated as follows, where $P(x)$ is the negative sampling distribution that is positively correlated with the degree of node $x$.

$$L_{tem} = -\log \sigma(\lambda_{(x,y,t)}) - \sum_{n \sim P(x)} \log \sigma(1 - \lambda_{(x,n,t)}) \tag{4}$$

For the above temporal information modeling, we introduce a classic HTNE method as the baseline without any additional changes. But the use of time alone is not enough to improve the performance of temporal graph clustering, so we propose clustering loss to compensate for this.

## 2.3 Improved Clustering Loss

Compared with static graph clustering, temporal graph clustering faces the challenge that deep clustering modules based on static adjacency matrix are no longer applicable. Since temporal graph methods train data in batches, we propose two new batch-based modules for node clustering, i.e., node-level distribution and batch-level reconstruction.

### 2.3.1 Node-Level Distribution

In this module, we focus on assigning all nodes to different clusters. Especially, for each node $x$ and each cluster $c_k$, we utilize the Student's t-distribution (Van der Maaten & Hinton, 2008) to measure their similarity.

$$q_{(x,k,t)} = \frac{(1 + ||\boldsymbol{z}_x^0 - \boldsymbol{z}_{c_k}^t||^2/v)^{-\frac{v+1}{2}}}{\sum_{c_j \in C}(1 + ||\boldsymbol{z}_x^0 - \boldsymbol{z}_{c_j}^t||^2/v)^{-\frac{v+1}{2}}} \tag{5}$$

Here $q_{(x,k,t)}$ denotes the probability of assigning node $x$ to cluster $c_k$, and $v$ is the degrees of freedom (default value is 1) for Student's t-distribution (Bo et al., 2020). $\boldsymbol{z}_x^0$ denotes the initial feature of node $x$, $\boldsymbol{z}_{c_k}^t$ is one clustering center embedding initialized by K-means on initial node features.

Considering that $q_{(x,k,t)}$ and $\boldsymbol{z}_x^0$ need to be as reliable as possible, and not all temporal graph datasets provide original features to ensure such reliability. We select the classical method node2vec (Grover & Leskovec, 2016) to generate initial features for nodes by mining the graph structure, which is equivalent to the pre-training. Note that many graph clustering methods use classical model pre-train to generate initialized clustering centers, such as SDCN (Bo et al., 2020) utilizes AE, DFCN (Tu et al., 2021) and DCRN (Liu et al., 2022a) utilize GAE, etc. Our selection of node2vec is not deliberate, and can be replaced by any other methods.

After calculating the initial probability, we aim to optimize the node embeddings by learning from the high-confidence assignments. In particular, we encourage all nodes to get closer to cluster centers, thus the target distribution $p_{(x,k,t)}$ at time $t$ can be sharpened as follows.

$$p_{(x,k,t)} = \frac{q_{(x,k,t)}^2/\sum_{i \in V} q_{(i,k,t)}}{\sum_{c_j \in C}(q_{(x,j,t)}^2/\sum_{i \in V} q_{(i,j,t)})} \tag{6}$$

The target distribution squares and normalizes each node-cluster pair in the assignment distribution to encourage the assignments to have higher confidence, then we can consider it as the "correct" distribution. We introduce the KL divergence (Kullback & Leibler, 1951) to the node-level distribution loss, where the real-time assignment distribution is aligned with the target distribution.

$$L_{node} = \sum_{c_k \in C} p_{(x,k,t)} \log \frac{p_{(x,k,t)}}{q'_{(x,k,t)}} \tag{7}$$

Note that $q'_{(x,k,t)}$ is calculated from node embeddings and can change with the update of node embeddings. This loss function aims to encourage the real-time assignment distribution as close as possible to the target distribution, so that the node embeddings can be more suitable for clustering.

The calculation of the assignment distribution differs in the static and temporal graphs. In static graph clustering, all nodes are computed simultaneously for the distribution. However, there is a sequential order of interactions in temporal graphs, so we calculate the distribution of the nodes in each interaction by batches. Note that if a node has multiple interactions, its distribution will be calculated multiple times, which we consider as multiple calibrations for important nodes.

### 2.3.2 Batch-Level Reconstruction

Graph reconstruction also plays an important role in node clustering, which can be considered as the pretext task. Due to the batch training in temporal graph learning, the adjacency matrix-based reconstruction technology can hardly be applied in temporal methods. Thus we propose the batch-level module to simply simulate the adjacency relationships reconstruction.

As mentioned above, for each batch, we calculate the temporal conditional intensity between nodes $x$ and $y$. To achieve that, we obtain the historical sequence $N_x$ of $x$. It means that both target node $y$ and neighbor nodes $h \in N_x$ have edges with $x$ in the graph, i.e., their adjacency relations are all 1. In addition, in the temporal loss function (Eq. 4), we also sample some negative nodes $n \sim P(x)$, which have no edges with $x$ in the real graph, i.e., their adjacency relations are all 0.

Based on the adjacency relationships above, the embedding of these nodes should also follow this constraint. Thus we utilize the cosine similarity to measure the relationship between two node embeddings and constrain them as close to 1 or 0 as possible. The cosine similarity between two node embeddings can be calculated as $\cos(\boldsymbol{z}_x, \boldsymbol{z}_y) = \frac{\boldsymbol{z}_x^\top \boldsymbol{z}_y}{||\boldsymbol{z}_x|| \cdot ||\boldsymbol{z}_y||}$.

This pseudo-reconstruction operation on batches, while not fully restoring the adjacency matrix reconstruction, uses as many nodes as possible that appear in the batch. It is equivalent to a simple reconstruction of the adjacency matrix without increasing the time complexity, which may provide a new idea for the problem that there is no adjacency matrix in batch processing of temporal graphs. Finally, the batch-level loss function can be formulated as follows.

$$L_{batch} = |1 - \cos(\boldsymbol{z}_x^t, \boldsymbol{z}_y^t)| + |1 - \cos(\boldsymbol{z}_x^t, \boldsymbol{z}_h^t)| + |0 - \cos(\boldsymbol{z}_x^t, \boldsymbol{z}_n^t)| \tag{8}$$

Thus the improved clustering loss function can be formulated as $L_{clu} = L_{node} + L_{batch}$.

## 2.4 LOSS FUNCTION AND COMPLEXITY ANALYSIS

Our total loss function includes temporal loss and clustering loss, which can be formulated as $L = \sum^E (L_{tem} + L_{clu})$. Note that the temporal graph is trained in batches, and this division of batches is not related to the number of nodes $N$, but to the length of the interaction sequence $|E|$ (i.e., the total number of interactions). This means that the main complexity of the method for temporal graph clustering is $O(|E|)$, rather than $O(N^2)$ for static graph clustering because the temporal graph method does not need to call the whole adjacency matrix. In other words, compared to static clustering, temporal clustering has the advantage of being more flexible and convenient for training:

(1) In the vast majority of cases, $|E|$ is smaller than $N^2$ because the upper bound of $|E|$ is $N^2$ (when the graph is fully connected), which means that $O(|E|) < O(N^2)$ in most time.

(2) In individual cases, there is a case where $O(|E|) > O(N^2)$, which means that there are multiple interactions between a large number of node pairs. This underscores the superiority of dynamic interaction sequences over adjacency matrix, as the latter compresses multiple interactions into a single edge, leading to a significant loss of information.

(3) The above discussion applies not only to the time complexity but also to the space complexity. As the interaction sequence is arranged chronologically, it can be partitioned into multiple batches of varying sizes. The maximum batch size is primarily determined by the available memory of the deployed platform and can be dynamically adjusted to match the memory constraints. Therefore, TGC can be deployed on many platforms without strict memory requirements.

The different types of datasets mentioned above are all considered in our experiments, which have very different node degrees and sizes. Then, We conduct experiments and discussions around these datasets from multiple domains.

## 3 DATASETS

A factor limiting the development of temporal graph clustering is that it is difficult to find a dataset suitable for clustering. Although node clustering is an unsupervised task, we need to use node labels when verifying the experimental results. Most public temporal graph datasets suffer from the following problems: (1) Researchers mainly focus on link prediction without node labels. Thus many public datasets have no labels (such as Ubuntu, Math, Email, and Cloud). (2) Some datasets have only two labels (0 and 1), i.e., models on these datasets aim to predict whether a node is active at a certain timestamp. Node classification tasks on these datasets tend to be more binary-classification than multi-classification, thus these datasets are also not suitable for clustering tasks (such as Wiki, CollegeMsg, and Reddit). (3) Some datasets' labels do not match their own characteristics, e.g., different ratings of products by users can be considered as labels, but it is difficult to say that these

Table 1: Dataset statistics.

| Datasets | Nodes | Interactions | Edges | Complexity | Timestamps | $K$ | Degree | MinI | MaxI |
|---|---|---|---|---|---|---|---|---|---|
| DBLP | 28,085 | 236,894 | 162,441 | $N^2 \gg E$ | 27 | 10 | 16.87 | 1 | 955 |
| Brain | 5,000 | 1,955,488 | 1,751,910 | $N^2 > E$ | 12 | 10 | 782 | 484 | 1,456 |
| Patent | 12,214 | 41,916 | 41,915 | $N^2 \gg E$ | 891 | 6 | 6.86 | 1 | 789 |
| School | 327 | 188,508 | 5,802 | $N^2 < E$ | 7,375 | 9 | 1153 | 7 | 4,647 |
| arXivAI | 69,854 | 699,206 | 699,198 | $N^2 \gg E$ | 27 | 5 | 20.02 | 1 | 11,594 |
| arXivCS | 169,343 | 1,166,243 | 1,166,237 | $N^2 \gg E$ | 29 | 40 | 13.77 | 1 | 13,161 |

labels are more relevant to the product characteristics than the product category labels, thus leading to the poor performance of all methods on these datasets (such as Bitcoin, ML1M, Yelp, and Amazon).

Constrained by these problems, as shown in Table 1, we select these suitable datasets from 40+ datasets: **DBLP** (Zuo et al., 2018) is a co-author graph from the DBLP website, which contains 10 research areas, i.e., 10 clusters. Each researcher is considered a node, and the collaborative relationships between them are considered interactions. **Brain** (Preti et al., 2017) is a human brain tissue connectivity graph, where the nodes represent tidy cubes of brain tissue, and the edges indicate connectivity. **Patent** (Hall et al., 2001) is a patent citation graph of US patents. Each patent belongs to six different types of patent categories. **School** (Mastrandrea et al., 2015) is a high school dataset that records the contact and friendship between school students. Although the number of students is small, they have many interactions throughout the day.

**arXivAI** and **arXivCS** (Wang et al., 2020) are two public citation graphs from the arXiv website, where papers are nodes, and citations are interactions. Note that these two datasets are developed by us large-scale temporal graph clustering, which record the academic citations on the arxiv website. Their original data are from the OGB benchmark (Wang et al., 2020), but are not applicable to temporal graph clustering. We extracted reference records from the original data to construct node interactions with timestamps and then find the corresponding node ids to construct the interaction sequence-style temporal graph.

To generate node labels suitable for clustering, we select the domain to which the paper belongs as its node label. Specifically, the arXiv website categorizes computer domains into 40 categories. We first identify the domains that correspond to the nodes, and then convert them into node labels. On the basis of arXivCS, we also construct the arXivAI dataset by extracting Top-5 relevant domains to AI from the original 40 domains and used them as the basis to extract the corresponding nodes and interactions.

In Table 1, *Nodes* denotes the node number, and *Interactions* denotes the interaction number. Note that we also report *Edges*, which represents the edge number in the adjacency matrix when we compress temporal graphs into static graphs for traditional graph clustering methods (As mentioned in Fig. 1, some duplicate interactions are missing). *Complexity* denotes the main complexity comparison between static graph clustering and temporal graph clustering, *Timestamps* means interaction time, $K$ means the number of clusters (label categories), and *Degree* means node average degree. *MinI* and *MaxI* denote the maximum and minimum interaction times of nodes, respectively.

## 4  EXPERIMENTS

In this part, we discuss the experiment results. *Due to the limitation of space, we present some of the descriptions and experiments in the appendix.* Here we ask several important questions about the experiment: **Q1**: What are the advantages of TGC? **Q2**: Is the memory requirement for TGC really lower? **Q3**: Is TGC valid for existing temporal graph learning methods? **Q4**: What restricts the development of temporal graph clustering?

### 4.1  BASELINES

To demonstrate the performance of TGC, we compare it with multiple state-of-the-art methods as baselines. In particular, we divide these methods into three categories: **Classic methods** refer to some early and highly influential methods, such as DeepWalk (Perozzi et al., 2014), AE (Hinton & Salakhutdinov, 2006), node2vec (Grover & Leskovec, 2016), GAE (Kipf & Welling, 2016), etc. **Deep graph clustering methods** refer to some methods that focus on clustering nodes on static graphs, such as MVGRL (Hassani & Khasahmadi, 2020), AGE (Cui et al., 2020), DAEGC (Wang et al., 2019), SDCN and SDCNQ (Bo et al., 2020), DFCN (Tu et al., 2021), etc. **Temporal graph**

Table 2: Node clustering results in common datasets. We bold the best results and underline the second best results. If a method face the out-of-memory problem, we record as OOM.

| Data | Metric | deepwalk | AE | node2vec | GAE | MVGRL | AGE | DAEGC | SDCN | SDCNQ | DFCN | HTNE | TGAT | JODIE | TGN | TREND | TGC |
|---|---|---|---|---|---|---|---|---|---|---|---|---|---|---|---|---|---|
| DBLP | ACC | 45.07 | 42.16 | 46.31 | 39.31 | 28.95 | OOM | OOM | 46.69 | 40.47 | 41.97 | 45.74 | 36.76 | 20.79 | 19.78 | _46.82_ | **48.75** |
| | NMI | 31.46 | 36.71 | 34.87 | 29.75 | 22.03 | OOM | OOM | 35.07 | 31.86 | _36.94_ | 35.95 | 28.98 | 1.70 | 9.82 | 36.56 | **37.08** |
| | ARI | 17.89 | 22.54 | 20.40 | 17.17 | 13.73 | OOM | OOM | **23.74** | 19.80 | 21.46 | 22.13 | 17.64 | 1.64 | 5.46 | 22.83 | _22.86_ |
| | F1 | 38.56 | 37.84 | 43.35 | 35.04 | 24.79 | OOM | OOM | 40.31 | 35.18 | 35.97 | 43.98 | 34.22 | 13.23 | 10.66 | _44.54_ | **45.03** |
| Brain | ACC | 41.28 | 43.48 | 43.92 | 31.22 | 15.76 | 38.48 | 42.52 | **47.46** | 43.20 | 43.42 | 43.20 | 41.43 | 19.14 | 17.40 | 39.83 | _44.30_ |
| | NMI | 49.09 | _50.49_ | 45.96 | 32.23 | 21.15 | 39.64 | 49.86 | 46.61 | 47.40 | 48.53 | 50.33 | 48.72 | 10.50 | 8.04 | 45.64 | **50.68** |
| | ARI | 28.40 | _29.78_ | 26.08 | 14.97 | 9.77 | 28.82 | 27.47 | 27.93 | 27.69 | 28.58 | 29.26 | 23.64 | 5.00 | 4.56 | 22.82 | **30.03** |
| | F1 | 42.54 | 43.26 | _46.61_ | 34.11 | 13.56 | 36.47 | 43.24 | 41.42 | 37.27 | **50.45** | 43.85 | 41.13 | 11.12 | 13.49 | 33.67 | 44.42 |
| Patent | ACC | 38.69 | 30.81 | 40.36 | 39.65 | 31.13 | 43.28 | _46.64_ | 37.28 | 32.76 | 39.23 | 45.07 | 38.26 | 30.82 | 38.77 | 38.72 | **50.36** |
| | NMI | 22.71 | 8.76 | _24.84_ | 17.73 | 10.19 | 20.72 | 21.28 | 13.17 | 9.11 | 15.42 | 20.77 | 19.74 | 9.55 | 8.24 | 14.44 | **25.04** |
| | ARI | 10.32 | 7.43 | 18.95 | 13.61 | 10.26 | **19.23** | 16.74 | 10.12 | 7.84 | 12.24 | 10.69 | 13.31 | 7.46 | 6.01 | 13.45 | _18.81_ |
| | F1 | 31.48 | 26.65 | 34.97 | 30.95 | 18.06 | _35.45_ | 32.83 | 31.38 | 28.27 | 30.32 | 28.85 | 26.97 | 20.83 | 21.40 | 28.41 | **38.69** |
| School | ACC | 90.60 | 30.88 | 91.56 | 85.62 | 32.37 | 84.71 | 34.25 | 43.94 | 49.85 | | _99.38_ | 80.54 | 19.88 | 31.71 | 94.18 | **99.69** |
| | NMI | 91.72 | 21.42 | 92.63 | 89.41 | 31.23 | 81.51 | 29.53 | 53.35 | 25.79 | 43.37 | _98.73_ | 73.25 | 9.26 | 19.45 | 89.55 | **99.36** |
| | ARI | 89.66 | 12.04 | 90.25 | 83.09 | 25.00 | 70.24 | 15.38 | 33.81 | 15.82 | 28.31 | _98.70_ | 80.04 | 2.85 | 32.12 | 87.50 | **99.33** |
| | F1 | 92.63 | 31.00 | 91.74 | 82.64 | 24.41 | 84.80 | 31.39 | 45.62 | 33.25 | 47.05 | _99.34_ | 79.56 | 13.02 | 29.50 | 94.18 | **99.69** |

Table 3: Node clustering results in large-scale datasets. We bold the best results and underline the second best results. If a method face the out-of-memory problem, we record as OOM.

| Data | Metric | deepwalk | AE | node2vec | GAE | MVGRL | AGE | DAEGC | SDCN | SDCNQ | DFCN | HTNE | TGAT | JODIE | TGN | TREND | TGC |
|---|---|---|---|---|---|---|---|---|---|---|---|---|---|---|---|---|---|
| arXivAI | ACC | 60.91 | 23.85 | 65.01 | 38.72 | OOM | OOM | OOM | 44.44 | 37.62 | OOM | _65.66_ | 48.69 | 30.71 | 31.25 | 29.82 | **73.59** |
| | NMI | 34.34 | 10.20 | 36.18 | 32.54 | OOM | OOM | OOM | 21.63 | 20.73 | OOM | _39.24_ | 32.12 | 2.91 | 24.74 | 1.28 | **42.46** |
| | ARI | 36.08 | 14.00 | 40.35 | 32.98 | OOM | OOM | OOM | 23.43 | 21.29 | OOM | _43.73_ | 30.34 | 5.35 | 11.91 | 1.12 | **48.98** |
| | F1 | 49.47 | 19.20 | _53.66_ | 16.97 | OOM | OOM | OOM | 33.96 | 31.62 | OOM | 52.86 | 43.62 | 23.24 | 21.93 | 19.22 | **57.86** |
| arXivCS | ACC | _34.42_ | 24.20 | 27.39 | OOM | OOM | OOM | OOM | 29.78 | 27.05 | OOM | 25.57 | 20.53 | 11.27 | 20.10 | 8.94 | **39.95** |
| | NMI | 40.86 | 14.03 | _41.18_ | OOM | OOM | OOM | OOM | 13.27 | 11.57 | OOM | 40.83 | 38.64 | 5.12 | 16.21 | 5.57 | **43.89** |
| | ARI | _24.65_ | 11.80 | 19.14 | OOM | OOM | OOM | OOM | 14.32 | 12.02 | OOM | 16.51 | 15.54 | 5.31 | 18.63 | 3.49 | **36.06** |
| | F1 | 20.39 | 12.33 | 21.41 | OOM | OOM | OOM | OOM | 14.08 | 13.28 | OOM | 19.56 | 13.23 | 4.85 | _22.67_ | 4.02 | **25.46** |

**learning methods** refer to some methods that model temporal graphs without node clustering task, such as HTNE (Zuo et al., 2018), TGAT (Xu et al., 2020), JODIE (Kumar et al., 2019), TGN (Rossi et al., 2020), TREND (Wen & Fang, 2022), etc.

## 4.2 Node Clustering Performance

**Q1**: *What are the advantages of TGC?* **Answer**: *TGC is more adapted to high overlapping graphs and large-scale graphs.* As shown in Table 2 and 3, we can observe that:

(1) Although TGC may not perform optimally on all datasets, the aggregate results are leading. Especially on large-scale datasets, many static clustering methods face the out-of-memory (OOM) problem on GPU (we use NVIDIA RTX 3070 Ti), only SDCN benefits from a simpler architecture and can run on the CPU (not GPU). This is due to the overflow of adjacency matrix computation caused by the excessive number of nodes, and of course, the problem can be avoided by cutting subgraphs. Nevertheless, we also wish to point out that by training the graph in batches, temporal graph learning can naturally avoid the OOM problem. This in turn implies that temporal graph clustering is more flexible than static graph clustering on large-scale temporal graph datasets.

(2) The performance varies between different datasets, which we believe is due to the datasets belonging to different fields. For example, the arXivCS dataset has 40 domains and some of which overlap, thus it is difficult to say that each category is distinctly different, so the performance is relatively low. On the contrary, node labels of the School dataset come from the class and gender that students are divided into. Students in the same class or sex often interact more frequently, which enables most methods to distinguish them clearly. Note that on the School dataset, almost all temporal methods achieve better performance than static methods. This echoes the complexity problem we analyzed above, as the only dataset where $N^2 < E$, the dataset loses the vast majority of edges when we transfer it to the adjacency matrix, thus static methods face a large loss of valid information.

(3) The slightly poor performance of temporal graph learning methods compared to deep graph clustering methods supports our claim that current temporal graph methods do not yet focus deeply on the clustering task. In addition, after considering the clustering task, TGC surpasses the static graph clustering method, also indicating that time information is indeed important and effective in the temporal graph clustering task. Note that these temporal methods achieve different results due to different settings, but we consider our TGC framework can effectively help them improve the clustering performance. Next, we will discuss TGC's memory usage and transferability.

## 4.3 GPU Memory Usage Study

**Q2**: *Is the memory requirement for TGC really lower?* **Answer**: *Compared to static graph clustering methods, TGC significantly reduces memory requirements.*

We first compare the memory usage of static clustering methods and TGC on different datasets, which we sort by the number of nodes. As shown in Fig. 2, we report their max memory usage. As the node number increases, the memory usages of static methods become larger and larger, and eventually out-of-memory problem occurs.

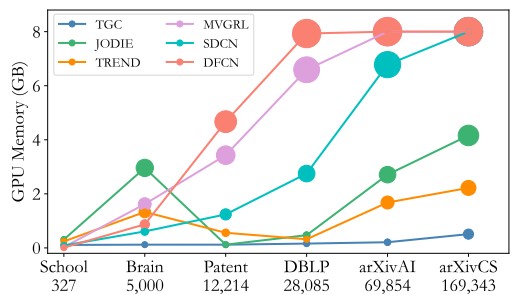

Figure 2: Memory changes between different datasets.

At the same time, the memory usage of TGC is also gradually increasing, but the magnitude is small and it is still far from the OOM problem. As mentioned above, the main complexity $O(|E|)$ of temporal methods is usually smaller than $O(N^2)$ of static methods. To name a few, for the arXivCS dataset with a large number of nodes and interactions, the memory usage of TGC is only 212.77 MB, while the memory usage of SDCN, the only one without OOM problem, is 6946.73 MB. For the Brain dataset with a small number of nodes and a large number of interactions, the memory usage of TGC (121.35 MB) is still smaller than SDCN (626.20 MB). We consider that for large-scale temporal graphs, TGC can be less concerned with memory usage, and thus can be deployed more flexibly to different platforms.

We further verify that TGC can flexibly adjust the batch size according to the memory. As we can see in Fig. 3, the runtime and memory footprint are basically inversely proportional on different timing diagram methods. This can also prove our conclusion that temporal graph clustering is able to find a balance between space requirements and time requirements. The only problem that occurs is the running time of TREND when the batch size is 10000, which increases rather than decreases. This is because the TREND model is designed to search for higher-order neighbors, a process that is done by the CPU. When the batch is larger, the number of nodes that have to wait for the CPU search result is also larger. Therefore, the increase in TREND's runtime on large batches is actually due to the increased computation by the CPU, rather than the GPU problem we usually consider. It also reflects the fact that TGC is flexible enough to find a balance between time consumption and space consumption according to actual requirements, either time for space or space for time is feasible.

## 4.4 TRANSFERABILITY AND LIMITATION DISCUSSION

**Q3**: *Is TGC valid for existing temporal graph learning methods?* **Answer**: *TGC is a simple general framework that can improve the clustering performance of existing temporal graph learning methods.*

As shown in Fig. 4, in addition to the baseline HTNE, we also add the TGC framework to TGN and TREND for comparison. Although TGC improves these methods differently, basically they are effective in improving their clustering performance. This means that TGC is a general framework that can be easily applied to different temporal graph methods.

We also want to ask **Q4**: *What restricts the development of temporal graph clustering?* **Answer**: *(1) Few available datasets and (2) information loss without adjacency matrix.*

On the one hand, as mentioned above, there are few available public datasets for temporal graph clustering. We collate a lot of raw data and transform them into temporal graphs, and we also discard many of them with low label confidence or incomplete labels. On the other hand, some global information is inevitably lost without adjacency matrix. Since temporal graph clustering is a novel task, there is still a lot of room for expansion of TGC. For example, how to further optimize module migration without adjacency matrix and how to adapt to the incomplete label problem in some graphs.

These issues limit the efficiency and performance of temporal graph clustering, and further exploration is required. In conclusion, although the development of temporal graph clustering is only in its infancy, we cannot ignore its possibility as a new solution.

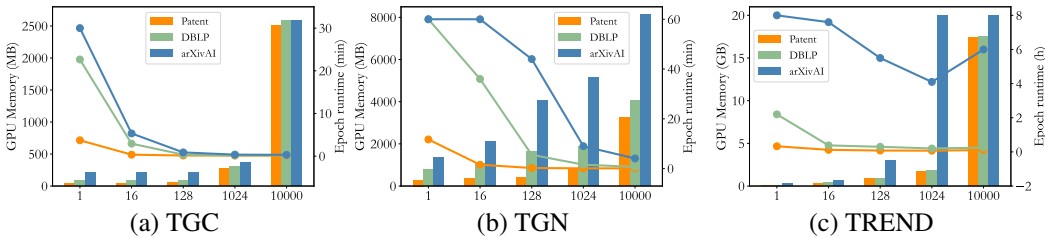

Figure 3: Changes in memory and runtime under different batch sizes.

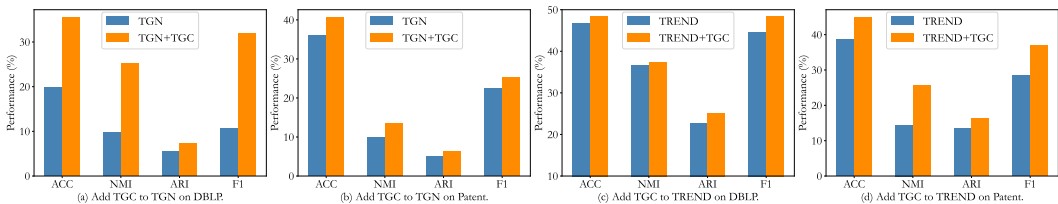

Figure 4: Transferability of TGC on different temporal graph methods.

Due to space constraints, we also present some of the content in the Appendix, which includes the Related Work section and more experiment details (i.e., experimental settings, ablation study, and parameter sensitivity study).

# 5 DISCUSSION

Here we make a summary statement in an attempt to bring out the focus of the paper:

(1) The main contribution of this paper is not only to present a generalized framework, but also to attempt to introduce and explain the direction of temporal graph clustering in several ways (intuition, method, data, experiment, etc.).

(2) We focus on temporal graph clustering because it offers a new possibility of clustering based on interaction sequences (adjacency lists) compared to clustering based on adjacency matrices in discrete dynamic or static graphs. Here, we group discrete dynamic and static graphs together because they both require the entire adjacency matrix to be read for training, which can pose a serious memory overflow problem.

(3) Interaction sequence-based temporal graph clustering does not suffer from memory overflow because it switches to a batch processing model. By feeding the interaction records of nodes into the model in batches, it is possible to modify the batch size to avoid exceeding the memory. We therefore point out that temporal graph clustering is able to flexibly find a balance between temporal and spatial demands, which certainly provides new ideas for current graph clustering models.

(4) Thus, what we want to show is not that temporal graph clustering performs better compared to other methods, but that it offers new horizons. Under this premise, even if there is a gap in the performance of temporal graph clustering compared to other methods, it does not detract from the benefits of its flexibility.

# 6 CONCLUSION

In this paper, we propose a general framework TGC for temporal graph clustering, which adapt clustering techniques to the interaction sequence-based batch-processing pattern of temporal graphs. To introduce temporal graph clustering as comprehensively as possible, we discuss the differences between temporal graph clustering and existing static graph clustering at several levels, including intuition, complexity, data, and experiments. Combining experiment results, we demonstrate the effectiveness of our TGC framework on existing temporal graph learning methods, and point out that temporal graph clustering enables more flexibility in finding a balance between time and space requirements. In the future, we will further focus on large-scale and $K$-free temporal graph clustering.

ACKNOWLEDGMENTS

This work was supported by the National Natural Science Foundation of China (Project No. 62325604, 62276271).

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

# A   RELATED WORK

## A.1   DEEP GRAPH CLUSTERING

Node clustering, commonly known as graph clustering, is a classic and crucial unsupervised task in graph learning (Hamilton, 2020; Liu et al., 2022b). Recently, the utilization of deep learning techniques for graph clustering has emerged as a research trend, resulting in the development of numerous methods (Liu et al., 2023c; Li et al., 2023; Mo et al., 2023a).

For instance, GraphEncoder (Tian et al., 2014) generates a non-linear embedding of the original graph through stacked autoencoders, followed by the K-means module to compute the clustering performance. DNGR (Cao et al., 2016) extracts a low-dimensional vector representation for every vertex, capturing the structural information of the graph. By utilizing an attention network to capture the importance of neighboring nodes to a target node, DAEGC (Wang et al., 2019) encodes the topological structure and node content of a graph into a compact representation. ARGA (Pan et al., 2018) encodes the graph structure and node information into a concise embedding, which is then trained using a decoder to reconstruct the structure. MVGRL (Hassani & Khasahmadi, 2020) is a self-supervised method that generates node representations by contrasting structural views of graphs. AGE (Cui et al., 2020) applies a Laplacian smoothing filter, carefully designed to enhance the filtered features for better node embeddings. SDCN (Bo et al., 2020) introduces a structural deep clustering network tointegrate the structural information into deep clustering. DFCN (Tu et al., 2021) utilizes two sub-networks to process augmented graphs independently. DCRN (Liu et al., 2022a) proposes a dual correlation reduction module to reduce information correlation in a dual manner. CGC (Park et al., 2022) learns node embeddings and cluster assignments in a contrastive graph learning framework, where positive and negative samples are carefully selected in a multi-level scheme.

The majority of these methods are based on static graphs, where different modules match the adjacency matrix, with the most common being the soft assignment distribution and graph reconstruction Wu et al. (2023; 2024). However, the deep clustering for temporal graphs remains largely unexplored. Temporal graphs emphasize time information between nodes in interaction sequence, which is an important data form of dynamic graphs.

## A.2   TEMPORAL GRAPH LEARNING

Graph data can be classified into static and dynamic graphs based on the presence or absence of time information Liang et al. (2022); Mo et al. (2023b). Traditional static graphs represent data in the form of an adjacency matrix, where samples are nodes and relationships between samples are edges (He et al., 2021). Various methods based on static graphs process the adjacency matrix in different ways to generate node embeddings, such as DeepWalk (Perozzi et al., 2014), which employs random walk to generate node representations. Node2vec (Grover & Leskovec, 2016) extends the random walk strategy to depth-first and breadth-first. AE (Hinton & Salakhutdinov, 2006) first proposes the auto-encoder framework, while GAE and VGAE (Kipf & Welling, 2016) further extend AE to graph data. GraphSAGE (Hamilton et al., 2017) is the first inductive method on graphs.

Dynamic graphs can be further classified into discrete graphs (discrete-time dynamic graphs, DTDGs) and temporal graphs (continuous-time dynamic graphs, CTDGs). Discrete graphs generate multiple static snapshots based on the fixed time interval, where each snapshot is considered as a static graph, and all snapshots are sorted chronologically (Gao & Ribeiro, 2021; Wu et al., 2021; Liang et al., 2023b). Discrete graph methods typically use the static model to learn each snapshot and then introduce RNN or attention modules to capturethe time information between different snapshots, such as EvolveGCN (Pareja et al., 2020) and DySAT (Sankar et al., 2020). In this case, discrete graphs can still be handled with common graph clustering technologies.

Unlike discrete graphs, temporal graphs can observe each node interaction more clearly. Temporal graphs, also known as continuous-time dynamic graphs (CTDGs), discard the adjacency matrix form and record node interactions based on the sequence directly. Temporal graph methods can divide data into batches and then feed it into the model for a single batch. For instance, HTNE (Zuo et al., 2018) employs the Hawkes process (Hawkes, 1971) to model historical neighbors' influence. JODIE (Kumar et al., 2019) aims to predict future node embeddings with uncertain time intervals. DyRep (Trivedi et al., 2019) combines local propagation, self-propagation, and external information patterns

to generate node embeddings. TGAT (Xu et al., 2020) encodes the time information and introduces the kernel function to decode it. TGN (Rossi et al., 2020) stores the historical memory of each node and then updates them after interactions. MNCI (Liu & Liu, 2021) considers both community influence and neighborhood influence to generate node representations inductively. TREND (Wen & Fang, 2022) utilizes a graph neural network module to model the conditional intensity between nodes. TMac (Liu et al., 2023a) introduces the multi-modal temporal graph network for audiovisual event classification.

### A.3 DIFFERENCE DISCUSSION

In fact, most temporal graph methods focus on link prediction rather than node clustering, which we attribute to the facts: On the one hand, temporal graph datasets rarely have corresponding node labels, and their data types are more suitable for targeting edges rather than nodes. On the other hand, the existing clustering techniques need to be adapted for the temporal graphs. Although there are very few methods that refer to the concept of temporal graph clustering from different perspectives, we should still point out that their description of temporal graph clustering is not sufficient:

(1) CGC (Park et al., 2022) claims to conduct the experiment of temporal graph clustering, but in fact, the experiment is based on discrete dynamic graphs and only carries out on one dataset. We acknowledge that discrete graphs and temporal graphs are inter-convertible, but temporal graphs have a more granular way of observing data than discrete graphs. Discrete graphs have to be processed as static graphs for each snapshot, which means that it can hardly process large-scale graphs. In other words, even discrete dynamic graphs (equivalent to static graphs) with only one static snapshot can cause memory overflow problems, so discrete graphs with multiple static snapshots combined will only appear to be more problematic in this regard. Thus we cannot give up the novel data processing pattern of temporal graph methods, which is one of the implications of temporal graph clustering. As the same reason, DNE (Du et al., 2018), RTSC (You et al., 2021), DyGCN (Cui et al., 2021), and VGRGMM (Li et al., 2022) are successful on discrete graphs, but not applicable to temporal graphs.

(2) GRACE (Yang et al., 2017) is a classical graph clustering method. Although its title includes "dynamic embedding", it actually refers to dynamic self-adjustment, which is still a static graph. STAR (Xu et al., 2019) and above Yao et al. (Yao & Joe-Wong, 2021) focus on node classification in temporal graphs, which is a mismatch with the node clustering task.

(3) Some methods claim to discuss clustering on dynamic or temporal graphs (Gorke et al., 2009; 2013; Matias & Miele, 2017; Nanayakkara et al., 2019; Ruan et al., 2021). However, they tend to use traditional machine learning or data mining algorithms rather than deep learning technologies, that we do not discuss here.

(4) Some works contain keywords such as "temporal / dynamic graph clustering" in the title but are less relevant to temporal graph. For example, DeGTeC (Liang et al., 2023c) is a data-parallel job framework for directed acyclic graphs, where the graph and temporal information are fully separated. DCFG (Bu et al., 2017) focuses on the dynamic cluster formation game in attribute graphs. DPC-DLP (Seyedi et al., 2019) considers the clustering task and uses KNN to propagate labels dynamically.

Through the introduction of some related work (more works will be described in Appendix), we consider that there is little comprehensive discussion of temporal graph clustering. With this motivation, we propose the general framework TGC and further discuss temporal graph clustering.

## B EXPERIMENTS

### B.1 EXPERIMENTAL SETTINGS

We conduct node clustering experiments on all datasets for all methods. First, we perform these models to generate node embeddings on all datasets, then utilize K-means to cluster these embeddings. We select Accuracy (ACC), Normalized Mutual Information (NMI), Average Rand Index (ARI), and macro F1-score (F1) as metrics.

We utilize Adam as the optimizer and select the value of hyper-parameters embedding dimension size $d$, batch size, historical sequence length $l$, negative sampling size $Q$, and learning rate as 128,

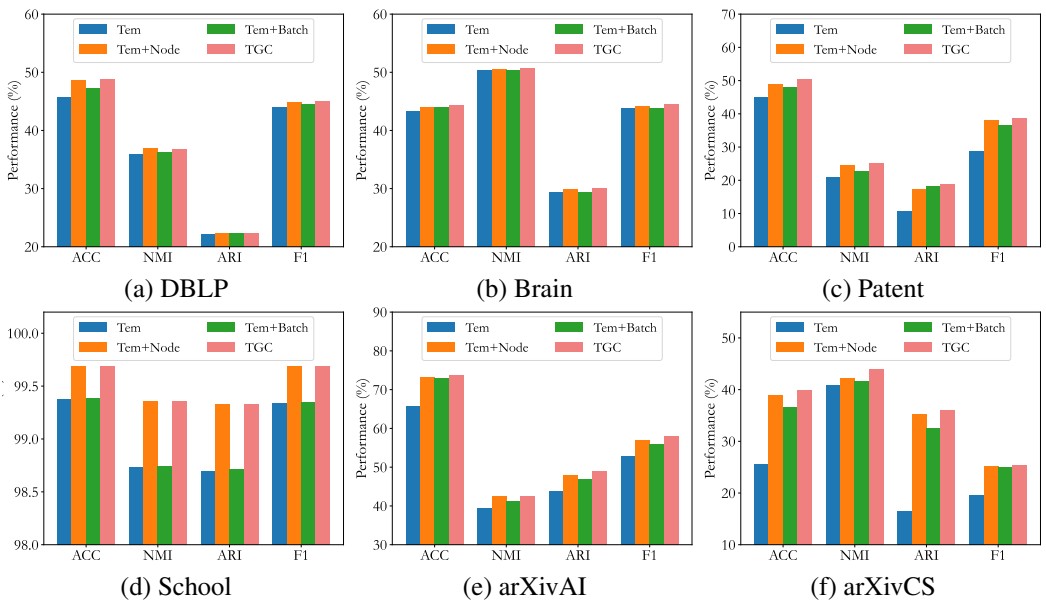

Figure 5: Ablation study on all datasets.

1024, 3, 5, and 0.01, respectively. We set the epoch number $T \leq 200$ on all datasets. For all baseline methods, we utilize their default parameter values.

Our proposed TGC framework is implemented with PyTorch, and all models are running on NVIDIA RTX 3070Ti GPUs (8GB), 64GB RAM, 3.2GHz Intel i9-12900KF CPU.

## B.2 ABLATION STUDY

To verify the effectiveness of our proposed module in TGC, we conducted an ablation study experiment on three datasets. Our TGC method includes temporal loss and clustering loss, where the temporal loss is a classic basic module introduced by many works and has not been modified. Therefore, the ablation study focuses only on the clustering loss, including the node-level module and batch-level module. We remove one module at a time to verify its effectiveness. Specifically, if the model only retains the temporal loss, we name it "**Tem**". If the model includes an additional node-level or batch-level module on top of the temporal loss, we respectively name them "**Tem+Node**" and "**Tem+Batch**". We compare these models with the full model "**TGC**".

According to Fig. 5, we observe that both the node-level and batch-level modules can effectively improve the performance of the model. In most cases, the node-level module has a greater effect on performance improvement than the batch-level module. The best performance is achieved when both modules are added to the model simultaneously. This suggests that the two modules we proposed can effectively enhance clustering performance, with the node-level module playing the most important role.

## B.3 PARAMETER SENSITIVITY STUDY

In this section, we analyze some hyper-parameters in TGC, including the historical sequence length, $l$, and the negative sampling number, $Q$.

As mentioned in the Problem Definition section, the historical sequence length, $l$, is an important hyper-parameter in temporal graph learning. In real-world graphs, the total number of neighbors may vary from node to node. To avoid computational inconvenience, we fix the sequence length, $l$, and select the latest $l$ neighbors for all nodes in each batch, instead of all neighbors. Based on previous works (Zuo et al., 2018; Lu et al., 2019; Xu et al., 2020; Wang et al., 2021; Wen & Fang, 2022; Liu et al., 2023b) and our experiments, we select different values for $l/Q$, i.e., $l/Q = 1/2/3/5/10$, to verify the sensitivity of $l$ and $Q$.

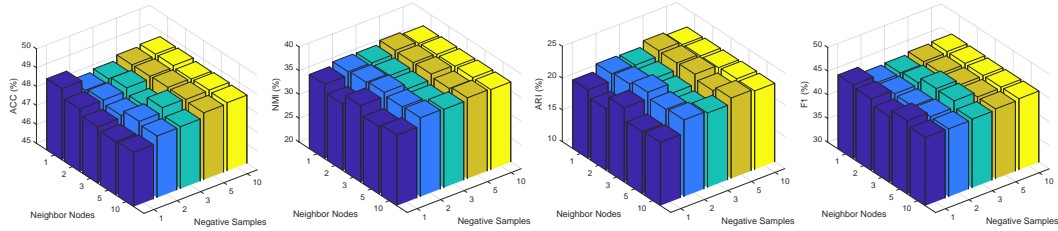

Figure 6: Parameter sensitivity study on the DBLP dataset.

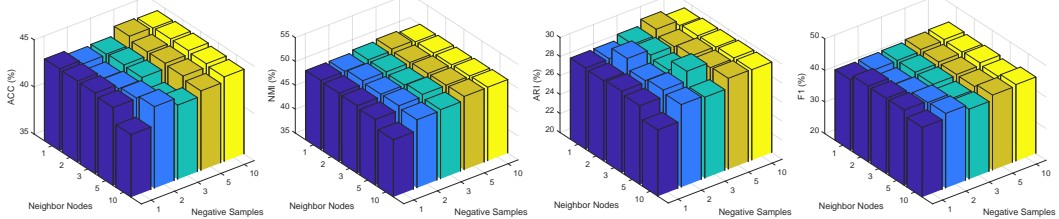

Figure 7: Parameter sensitivity study on the Brain dataset.

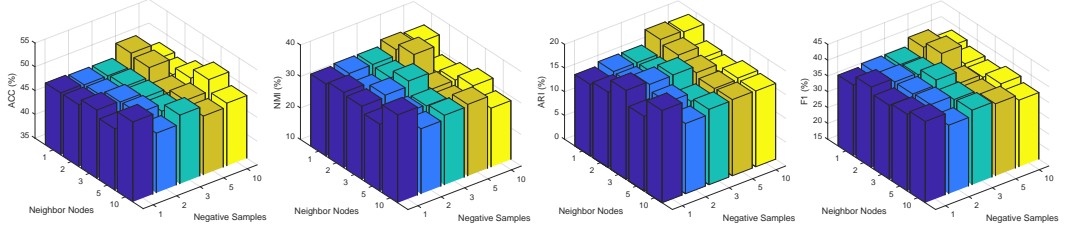

Figure 8: Parameter sensitivity study on the Patent dataset.

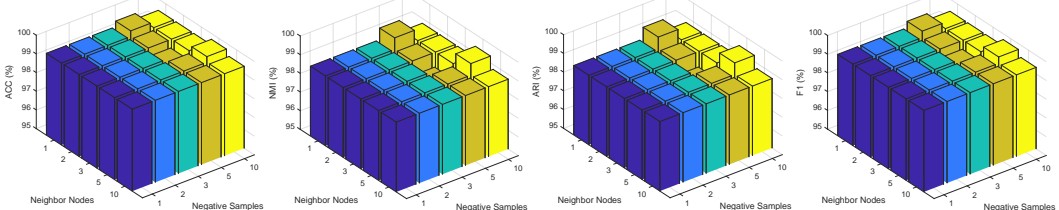

Figure 9: Parameter sensitivity study on the School dataset.

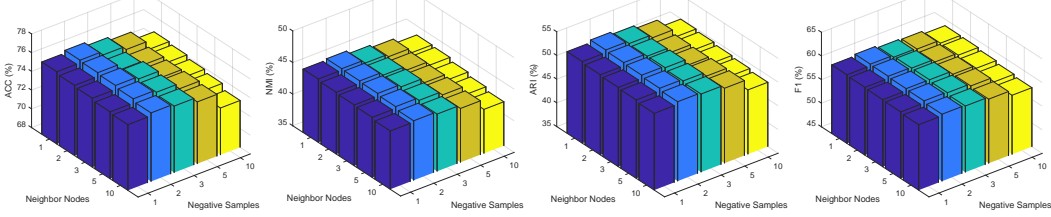

Figure 10: Parameter sensitivity study on the arXivAI dataset.

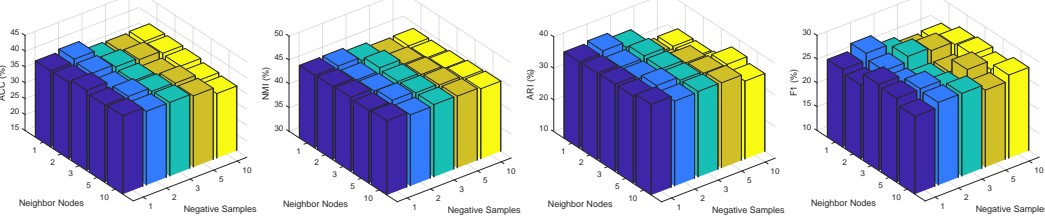

Figure 11: Parameter sensitivity study on the arXivCS dataset.

As shown in Figs. 6–11, we report the effects of different parameter settings. It can be found that:

(1) TGC is relatively insensitive to different parameter settings, as the experimental results often fluctuate within a small range. This demonstrates the stability of our framework, which is not constrained by hyper-parameter settings.

(2) Changing the parameters will inevitably have some impact on the experimental results, from which we can discover patterns. Regarding the historical neighbor sequence length, $l$, we find that larger values are not always better, which is consistent with our previous statement. When $l$ is too small, the model may not obtain enough neighborhood information, resulting in poor performance. As $l$ increases, the performance gradually improves, but after reaching a certain threshold, the performance decreases again. This is because an excessively long neighbor sequence considers very early neighbor nodes, which have little influence on the current interaction and instead introduce noise. This also reflects that time information is relatively important, and different considerations of time information will bring different changes in performance. In addition, we argue that the optimal length of $l$ varies on different datasets, depending on the average degree of the dataset.

(3) The selection of the number of negative samples, $Q$, for negative sampling has a high degree of uncertainty, which is not only related to the average node degree of different datasets but also affected by different model structures. Generally, choosing $Q = 2/3/5$ leads to better results, and the variation in $Q$ does not have a significant impact. This is because we encourage the positive conditional intensity to be large enough, while also restricting negative intensities to be as small as possible. The ideal case is that negative intensities go to 0. In other words, as all of these negative intensities go to 0, no amount of negative samples will make much difference to the loss value. Therefore, the low sensitivity of TGC to $Q$ precisely indicates that it can distinguish positive and negative samples well.

In summary, the TGC framework is not sensitive to the choice of hyper-parameters, and we set default values that can be used for all datasets, i.e., $l = 3$ and $Q = 5$, for convenience.

