# OpenReview forum: "Deep Temporal Graph Clustering"
_ICLR.cc/2024/Conference — ICLR 2024 poster_

### Official Review · Reviewer_PosK · 2023-10-20

**Soundness:** 3 good
**Presentation:** 4 excellent
**Contribution:** 4 excellent
**Rating:** 8
**Confidence:** 4

**Summary:**

This paper expands deep graph clustering to temporal graphs from four aspects, including problem, algorithm, dataset, and evaluation. A generalized framework named TGC is proposed for temporal graph clustering. The authors demonstrate the superiority of TGC and provide sound insights via extensive experiments.

**Strengths:**

+ The motivation is clear. It is meaningful to expand deep clustering to temporal graphs. The presentation is excellent.

+ The core idea is novel and easy to follow. The temporal loss mines graph temporal information and the clustering loss is improved with node-level distribution and batch-level reconstruction.

+ The experiments are comprehensive. The analyses provide many significant insights for temporal graph clustering.

**Weaknesses:**

- The authors just conduct the clustering algorithms one run. However, most clustering algorithms are not robust and are sensitive to random seeds.

- The performance of TGC on the Brain dataset is not promising. Please provide the explanation and analyses.

- The related work section is limited. The authors need to survey and compare more papers published in 2022 and 2023.

- Missing middle-scale datasets. The authors should explore the dataset size boundaries of the statics deep graph clustering methods.

- The chronological order training is a strong restriction. I’m concerned that such as this strong strict will limit the performance on some datasets. As I know, some methods can disrupt the graph structure for training.

- In Figure 11, point out which sub-figure represents the result on which dataset. Besides, ablation studies should be provided in the main text. Figure 2 is too small. In addition, there seems to be something wrong with the commas.

- Have you considered explaining the relationship between the number of nodes in the static graph and the number of edges in the temporal graph further? This is an exciting topic, i.e., what is the difference in the way of thinking about a problem when the focus shifts from nodes to edges?

**Questions:**

See Strengths and Weaknesses.

---

> ### Author Response · Authors · 2023-11-16
> **Response to Reviewer PosK [1/2]**
>
> We thank the Reviewer PosK for the detailed and constructive review. We greatly appreciate the valuable insights and extensive expertise reflected in your comments. Irrespective of the final acceptance of the article, we extend our sincerest gratitude for your suggestions, which have significantly enhanced the overall quality of our work. We respond to the comments one by one.
>
> > Due to space constraints, we have added the needed figures and tables to the last subsection (Page 21) of the appendix for your review. The rest of the text will be added after a subsequent reorganization of the article layout.
>
> ## **Experimental Repeatability**
>
> + This is an oversight in our presentation; our results are averaged over three experiments performed, not just one run. It is only due to restricted table space that we do not report the margin of error for each result.
>
> ## **Performance on Brain**
>
> + This can actually support our assertion that depth map clustering and temporal graph clustering are complementary and that they are applicable in different situations, rather than who is to replace whom.
>
> + For temporal graph data, when the number of nodes and edges is small, i.e., the graph scale is small, it is perfectly possible to learn by means of deep graph clustering. As in the case of the Brain dataset, deep graph clustering is able to obtain a more complete neighborhood structure, which is advantageous for clustering. In this case, temporal graph clustering may not be more effective than depth graph clustering.
>
> + However, when executed on large-scale datasets (e.g., arXIvCS), these depth graph clustering methods can suffer from OOM problems due to their inability to handle large-scale neighbor matrices. In this case, temporal graph clustering has its place.
>
> ## **Related Work**
>
> + Thanks to your reminder, we will introduce some relevant papers published in 2022 and 2023 in the Related Work Section [1-5]. Due to page constraints, we may consider including this section in detail in the appendix.
>
> > [1] Wang, Tianchun, et al. DyExplainer: Explainable Dynamic Graph Neural Networks. WSDM 2024.
>
> > [2] Feng, Kaituo, et al. Towards Open Temporal Graph Neural Networks. ICLR 2023.
>
> > [3] Liu, Meng, et al. TMac: Temporal Multi-Modal Graph Learning for Acoustic Event Classification. ACM MM 2023.
>
> > [4] Zheng, Yanping, et al. Decoupled Graph Neural Networks for Large Dynamic Graphs. VLDB 2023.
>
> > [5] Luo, Yuhong, et al. Neighborhood-aware Scalable Temporal Network Representation Learning. LOG 2022 Best Paper.
>
> ## **About Middle-Scale Datasets**
>
> + We should recognize that we have not consider the middle-scale datasets.  This is because it is already difficult to find public datasets suitable for temporal graph clustering, and even more difficult to satisfy the scale requirements. As shown in Table 4, we summarize some common public datasets, and we can see that these datasets can generally be classified into three categories.
>
> + The first category is datasets without usable labels (e.g., CollegeMsg), and in fact most of the common temporal graph datasets are of this type. The second category is those with untrustworthy labels (e.g., Bitcoin), and we have conducted experiments on datasets in this category, and their experimental results are often lower than 5\% or even 1\%, which means that it is difficult to measure the clustering performance with such node labels. The third category, i.e., the datasets we selected, may have some datasets with smaller sizes and some datasets with fewer timestamps, but these are already as suitable as possible for temporal graph clustering.
>
> + Therefore, in this case, it is difficult to find all the datasets that fulfill the different scales. In the future, we will consider designing and developing more temporal graph clustering datasets with different scales.

---

> ### Author Response · Authors · 2023-11-16
> **Response to Reviewer PosK [2/2]**
>
> ## **Training Order**
>
> + This is exactly the problem we wanted to point out, namely that the destruction of the graph structure has the potential to cause a loss of information about the graph structure. As we mentioned in the article: Suppose there is an interaction between two nodes in a subgraph occurs in 2023, while an edge between a subgraph and another So how do we reverse time the information from 2023 and pass it on in 2020?
>
> + In some scenarios where temporal information is more important, the destruction of the graph structure and thus the reverse propagation of information can instead negatively affect the clustering effect. Therefore, in our opinion, this strict time entry order is instead one of the advantages of temporal graph clustering, i.e., this guarantees a strict order of information transfer.
>
> ## **Figures and Ablation Study**
>
> + Regarding the ablation experiments of TGC node-level modules and batch-level modules, we provide the relevant experimental results in the Appendix section, as shown in Fig. 11. They are, in order, DBLP, Brain, Patent, School, arXivAI and arXivCS.
>
> + In addition, parameter sensitivity experiments on different datasets are provided in Fig. 5-Fig. 10. Due to space constraints, we did not put these experiments in the main text, after which we will consider adjusting the layout of the article. For the other issues you mentioned, we will amend them one by one, thank you for your correction.
>
> ## **Discussion from Node to Interaction**
>
> + You ask a very insightful question, which provokes us to think deeper: what exactly are we focusing on once the center of attention has shifted from nodes to edges?
>
> + First of all, the treatment of data becomes different, which is of course obvious, and we discuss this throughout. The fact that temporal graphs record data as a sequence of interactions and can thus be trained in batches brings more flexibility to the temporal graph approach, but at the same time many opportunities to use classical techniques are lost. Therefore we would like to emphasize that temporal graph clustering is not proposed to replace depth graph clustering on temporal graphs, it is proposed to provide more possibilities for researchers in. When researchers feel that static graph clustering is not very suitable, they can have one more option of temporal graph clustering.
>
> + In addition, the perspective of measuring the size of the dataset has changed. Static graphs judge the size of the graph by the number of nodes, while temporal graphs judge it by the number of interactions. This means that the two types of methods refer to different sizes when faced with large-scale problems. Although the number of nodes and the number of interactions tend to grow together, there are differences. For example, in several of the datasets we have given, the Brain dataset is clearly a large-scale temporal graph dataset.
>
> + Finally, we would like to point out to the readers that for temporal graph clustering, which is full of countless expandable spaces, this is still a near-new field. There are many classical techniques that need to be adapted to be applicable to temporal graphs, and this process of adaptation is the way to fully exploit temporal graph data.

---

> ### Comment · Reviewer_PosK · 2023-11-20
>
> The authors have addressed most of my concerns well. I think this is an innovative work in temporal graph clustering that can inspire other researchers in the community, so I recommend accepting this paper.

---

### Official Review · Reviewer_Euq6 · 2023-10-29

**Soundness:** 2 fair
**Presentation:** 2 fair
**Contribution:** 2 fair
**Rating:** 6
**Confidence:** 4

**Summary:**

This paper studies temporal graph clustering via graph neural networks. Temporal graphs have been widely investigated in the graph community. However, the topic of temporal graph clustering is less explored. The authors argue that this is because of lacking datasets suitable for clustering, thus several datasets are carefully selected by the authors for experiments. Moreover, the authors also introduce two tricks (i.e., clustering assignment distribution and adjacency matrix reconstruction) to improve existing models’ performance on temporal graph clustering task. Experimental results on 6 public datasets demonstrate that the proposed general framework TGC can outperform recent baselines with different gains.

**Strengths:**

1)	This paper investigates temporal graph clustering, which is less explored in the graph community.
2)	Both large and small datasets are used in the experimental sections and the results demonstrate the proposed model achieve really good performance.
3)	Memory consumption and running time are reported to show the efficiency of the proposed framework.
4)	Ablation studies are given to show the effectiveness of the proposed components.

**Weaknesses:**

1)	The technical contribution of this paper is very limited. The two introduced tricks (i.e., clustering assignment distribution and adjacency matrix reconstruction) have been widely used by existing works. Eq. 6 and Eq. 7 are commonly utilized in existing static graph clustering models. No new models are proposed in this paper.

[1] Bo D, Wang X, Shi C, et al. Structural deep clustering network[C]//Proceedings of the web conference 2020. 2020: 1400-1410.

[2] Xie J, Girshick R, Farhadi A. Unsupervised deep embedding for clustering analysis[C]//International conference on machine learning. PMLR, 2016: 478-487.

2)	This paper studies continuous-time dynamic graphs. The authors argue that their selected datasets are more suitable for temporal graph clustering tasks. However, it seems that it is not true. As we can see in Table 1, most of the selected datasets have very limited timestamps, which is not consistent with their claimed continuous-time dynamic graphs. They are more like discrete-time dynamic graphs. For example, the DBLP, arXivAI and arXivCS datasets usually contain publication year only.
3)	The comparisons of GPU memory usage are not fair in Figure 2. The authors only compare their TGC with static deep clustering baselines that feed whole graphs. The authors should compare their TGC with existing temporal graph clustering baselines like HTNE, TGAT, JODIE, etc. The low consumption of TGC largely depend on its backbone model HTNE, thus it is hardly to say this is the advantage of TGC. This comment can also be applied to Figure 3. The running time should also compare with other temporal baselines.
4)	In Figure 4, it is not clear what is TGN + TGC? Since TGC is based on HTNE, I guess what you mean is applying clustering assignment distribution and adjacency matrix reconstruction to TGN. If so, please give the ablation studies on each component.

There are lots of typos in the paper. Some of them are listed as follows:

“A sample general framework” should be “A simple general framework…”

“which we do not discuss here” should be “that we do not discuss here”

“Then We conduct” should be “Then, we conduct”

**Questions:**

1)	The comparisons of GPU memory usage are not fair in Figure 2. The authors only compare their TGC with static deep clustering baselines that feed whole graphs.
2)	The running time should also compare with other temporal baselines.
3)	It is not clear how clustering assignment distribution and adjacency matrix reconstruction will individually impact the model's performance.

---

> ### Author Response · Authors · 2023-11-16
> **Response to Reviewer Euq6 [1/3]**
>
> We thank the Reviewer Euq6 for the detailed and constructive review. We greatly appreciate the valuable insights and extensive expertise reflected in your comments. Irrespective of the final acceptance of the article, we extend our sincerest gratitude for your suggestions, which have significantly enhanced the overall quality of our work. We respond to the comments one by one.
>
> > Due to space constraints, we have added the needed figures and tables to the last subsection (Page 21) of the appendix for your review. The rest of the text will be added after a subsequent reorganization of the article layout.
>
> ## **Technical Contribution**
>
> Thank you for your comment. Before discussing this issue, we would like to reiterate our contributions.
>
> + (1) Our contributions can be divided into four parts. For **Problem**, We discuss the differences between temporal graph clustering and existing works on an intuitive level. For **Algorithm**, we design a general framework TGC, and analysis the computational complexity of temporal methods and static methods. For **Dataset**, we collate several suitable datasets and also develop two new datasets. For **Evaluation**, we conduct multiple experiments to elucidate the characteristics of temporal graph clustering.
>
> + (2) Then we would like to emphasize the point that model design is far from being the whole story of this article, and in terms of contributions it is only a quarter of the story. Moreover, even if we simply improve and apply these classical techniques, it is already the first time that they have been considered in temporal graph learning.
>
> + (3) This comes down to the reality that deep temporal graph clustering is a field that gets very little attention. In this case, much of the work we do is exploring uncharted territory. Within the paper, we expound upon deep temporal graph clustering, encompassing aspects such as intuition, modeling, complexity, datasets, and experiments. The deliberate selection of a simple model was intended to prevent an excessive focus on the model itself, allowing readers to grasp the broader concepts.
>
> + (4) Undoubtedly, we could have introduced various novel technical ideas, such as spiking Neural Network [1], parameter-free learning [2], coupled memory networks [3], open-world scenarios [4], and graph connectivity discussions [5]. These additions would certainly make our model appear to be sufficiently work-intensive and innovative, but we select not to do so, these techniques are beyond the focus of this paper. In the case of a nascent field, the employment of simple and easily comprehensible models is more conducive to its development. Subsequent researchers will naturally refine and enhance their models.
>
> + (5) For a paper about exploring a new field, too much technical detail can cloud the readers' judgment and attract too much attention, thus ignoring many details that we would like to discuss in other chapters. In our view, these details pertaining to the concept of temporal graph clustering hold greater significance than the model complex design. In other words, we constructed the model with the primary purpose of conducting experiments that further elucidate the characteristics of temporal graph clustering.
>
> Building upon this foundation, we have devised improvements to two classical techniques to render them adaptable to temporal graph data. Furthermore, we hope that the reviewers comprehend that, when exploring a new field, the majority of the workload lies not in the design of the model, but rather in problem definition, dataset selection, and experimental presentation.
>
> > [1] Li, Jintang, et al. Scaling Up Dynamic Graph Representation Learning via Spiking Neural Networks. AAAI 2023.
>
> > [2] Liu, Jiahao, et al. Parameter-free Dynamic Graph Embedding for Link Prediction. NeurIPS 2022.
>
> > [3] Zhang, Zhen, et al. Learning Temporal Interaction Graph Embedding via Coupled Memory Networks. WWW 2020.
>
> > [4] Feng, Kaituo, et al. Towards Open Temporal Graph Neural Networks. ICLR 2023.
>
> > [5] Zhang, Bohang, et al. Rethinking the Expressive Power of GNNs via Graph Biconnectivity. ICLR 2023 Outstanding Paper.

---

> ### Author Response · Authors · 2023-11-16
> **Response to Reviewer Euq6 [2/3]**
>
> ## **Timestamps in Datasets**
>
> We should recognize that these datasets have fewer timestamps. But they are already relatively suitable for temporal graph clustering in many public datasets. As shown in Table 4, we summarize some common public datasets, and we can see that these datasets can generally be classified into three categories.
>
> + (1) The first category is datasets without usable labels (e.g., CollegeMsg), and in fact most of the common temporal graph datasets are of this type. The second category is those with untrustworthy labels (e.g., Bitcoin), and we have conducted experiments on datasets in this category, and their experimental results are often lower than 5\% or even 1\%, which means that it is difficult to measure the clustering performance with such node labels. The third category, i.e., the datasets we selected, may have some datasets with smaller sizes and some datasets with fewer timestamps, but these are already as suitable as possible for temporal graph clustering.
>
> + (2) In addition, we would like to point out that many continuous-time dynamic graph (i.e., temporal graph) methods are also experimenting with these datasets with fewer timestamps [1-6], which may be unhelpful, but it does not affect the unfolding of the experiments on these datasets either. Since these methods focus on the mode of processing timestamps, and whatever the number of timestamps is, it does not affect the difference between the modes of processing of the continuous-time dynamic graphs and the discrete-time dynamic graphs.
>
> + (3) Going back to our dataset, we actually provide optional datasets for different scenarios. In terms of timestamp scale, there's Brain, which has fewer timestamps, Patent, which is medium-sized, and School, which has a lot of timestamps. These datasets can handle different situations, even those datasets with fewer timestamps. As we said, discrete dynamic graph methods usually use static graph clustering, which can cause OOM problems. If we converting discrete graphs to timestamped graphs for clustering, this can provides researchers with more ideas, and Brain can be seen as just such an example.
>
> Finally, in the future, we will aim to explore datasets with more timestamps.
>
> > [1] Feng, Kaituo, et al. Towards Open Temporal Graph Neural Networks. ICLR 2023.
>
> > [2] Xu, Da, et al. Inductive Representation Learning on Temporal Graphs. ICLR 2020.
>
> > [3] Rossi, Emanuele, et al. Temporal graph networks for deep learning on dynamic graphs. ICML Workshop 2020.
>
> > [4] Zhang, Zhen, et al. Learning temporal interaction graph embedding via coupled memory networks. WWW 2020.
>
> > [5] Kumar, Srijan, et al. Predicting dynamic embedding trajectory in temporal interaction networks. KDD 2019.
>
> > [6] Zuo, Yuan, et al. Embedding temporal network via neighborhood formation. KDD 2018.

---

> ### Author Response · Authors · 2023-11-16
> **Response to Reviewer Euq6 [3/3]**
>
> ## **Comparison with Temporal Baseline**
>
> Thanks to your reminder, we have added more comparison methods to both experiments on dataset size and batch size. We have placed the updated experimental plots on the last page of the Appendix (Figures 13 and 14) for your review.
>
> + (1) Specifically, HTNE has less GPU requirements due to its simplicity and flexibility, which is why we chose HTNE as the baseline model. In addition, JODIE and TREND will have a bit more GPU requirements, but still have a lot of upside compared to static graph clustering methods. Therefore, we believe that the batchwise temporal graph method is more suitable for large-scale temporal graph data. Moreover, as the data size increases, the temporal graph method can avoid the OOM problem by increasing the memory or shrinking the batch.
>
> + (2) This is exactly what we are going to discuss in the next experiment, i.e., memory and runtime changes due to batch changes. As we can see in Figure 14, the runtime and memory footprint are basically inversely proportional on different timing diagram methods. This can also prove our conclusion that temporal graph clustering is able to find a balance between space requirements and time requirements. The only problem that occurs is the running time of TREND when the batch size is 10000, which increases rather than decreases. This is because the TREND model is designed to search for higher-order neighbors, a process that is done by the CPU. When the batch is larger, the number of nodes that have to wait for the CPU search result is also larger. Therefore, the increase in TREND's runtime on large batches is actually due to the increased computation by the CPU, rather than the GPU problem we usually consider.
>
> + (3) In conjunction with the above, we should note that traditional graph clustering models tend to suffer from large-scale problems. And our proposed temporal graph clustering can bring new ideas to researchers, i.e., using the flexibility of temporal graphs to solve the scale problem of clustering, which is exactly what we want to emphasize.
>
> ## **Ablation Study**
>
> Thanks for the correction, here TGC refers to a clustering framework that can be flexibly adapted to other temporal graph methods and does not include HTNE. this is our mistake and we will correct this in the main text.
>
> + Regarding the ablation experiments of TGC node-level modules and batch-level modules, we provide the relevant experimental results in the Appendix section, as shown in Fig. 11. They are, in order, DBLP, Brain, Patent, School, arXivAI and arXivCS.
>
> + In addition, parameter sensitivity experiments on different datasets are provided in Fig. 5-Fig. 10. Due to space constraints, we did not put these experiments in the main text, after which we will consider adjusting the layout of the article.
>
> ## **Typos**
>
> Thank you for the corrections, we will correct these errors and recheck the full article. For example, some changes are given below:
>
> > (1) Change “A sample...” to “A simple...”.
>
> > (2) Change “which we do not...” to “that we do not...”.
>
> > (3) Change “Then We conduct” to “Then, we conduct”.
>
> > (4) Change “node embeding” to “node embeddings”.
>
> > (5) Corrected the mistake with the top quote.
>
> Thank you again for your professional comments and insights, which have helped us to improve the quality of our article.

---

> > ### Comment · Reviewer_Euq6 · 2023-11-22
> > **Thanks for your rebuttal!**
> >
> > I have read the authors' rebuttal. It almost addresses my concerns, thus I increase the score.

---

### Official Review · Reviewer_8jHD · 2023-10-30

**Soundness:** 3 good
**Presentation:** 3 good
**Contribution:** 4 excellent
**Rating:** 8
**Confidence:** 5

**Summary:**

This work focuses on deep temporal graph clustering and proposes a general framework for deep Temporal Graph Clustering called TGC, which introduces deep clustering techniques to suit the interaction sequence-based batch-processing pattern of temporal graphs. The experimental results are impressive and demonstrate the effectiveness of the proposed framework.

**Strengths:**

1 The paper is well-written, structured and easy to follow. The authors provide sufficient implementation and experimental details in the Appendix.
2 The authors' focus and exploration of temporal graph clustering may shed new light on the graph learning community.
3 The experiments are extensive and the results are presented in a clear and concise manner.

**Weaknesses:**

1 The authors mention that datasets are processed by batches, so for each batch does that mean it is a subgraph structure obtained by sampling? Or what is the difference between this way of processing the data in batches, compared to the way of training according to subgraphs?

2 I would like the authors to further discuss the practical application of temporal graph clustering in real-world scenarios to demonstrate the significance of this ‘new’ task.

3 The authors need to check the paper for grammatical and spelling errors, such as the wrong quotation marks above the Eq (7).

**Questions:**

Why isn't there more discussion about the two new datasets that you developed yourself in the paper? It doesn't seem to be presented very well in the appendix section either.

---

> ### Author Response · Authors · 2023-11-16
> **Response to Reviewer 8jHD [1/2]**
>
> We thank the Reviewer 8jHD for the detailed and constructive review. We greatly appreciate the valuable insights and extensive expertise reflected in your comments. Irrespective of the final acceptance of the article, we extend our sincerest gratitude for your suggestions, which have significantly enhanced the overall quality of our work. We respond to the comments one by one.
>
> > Due to space constraints, we have added the needed figures and tables to the last subsection (Page 21) of the appendix for your review. The rest of the text will be added after a subsequent reorganization of the article layout.
>
>
> ## **Batch Data**
> Thanks for the reviewer's comment, it was a good one and very insightful. It helped us reorganize our thinking and we take the opportunity to discuss it further here.
>
>
> + The temporal graph data is read in batches, and each batch holds the node interactions, i.e., each row considers only two nodes interacting at some moment. This means that different rows (i.e., different interactions) within the same batch may be completely unconnected to each other. Therefore, from this perspective, the interaction data of a single batch does not necessarily constitute a subgraph.
>
> + Further, it is precisely because a single batch cannot constitute a subgraph that the focus is on a single interaction and an attempt is made to obtain information about its neighbors for the nodes in the interaction, i.e., to construct a limited subgraph for a single node. This point is the biggest difference from those schemes based on batch subgraphs. This in turn leads to the fact that the temporal graph learning is not affected by the batch size in training, because even if there is only one interaction, the sequence of neighbors of a node is still constructed normally, and the batch size only affects the computational efficiency.
> + Subgraph-based methods, on the other hand, will have requirements on the batch size; when the batch is small, the subgraph structure that can be accessed lacks important information, and when the batch is too large, it will bring computational problems.
>
> Based on the above, we would like to reiterate once again that the batch processing mode based on interaction sequences for temporal graphs is not a replacement for any other algorithms (e.g., adjacency matrices, subgraphs, etc.), but rather it provides a new way of thinking outside of these algorithms, allowing for one more possibility of problem solving solutions.
>
>
> ## **Practical Application**
> Temporal graph clustering has a wide range of application scenarios. As the extension of graph clustering, theoretically all the real-life scenarios to which graph clustering applies, temporal graph clustering is applicable. Specifically, it can be divided into two categories, one is the application scenarios directly oriented to graph clustering, and the other is the application scenarios enhanced by graph clustering.
>
> + The first class of scenarios refers to the scenarios to which graph clustering tasks are naturally adapted, such as community discovery, anomaly detection, etc. The purpose of such scenarios is to mine latent relationships within the same cluster (community discovery) or anomalies that stray from the majority of clusters (anomaly detection) by categorizing the samples into different clusters, so the essence of these scenarios is precisely graph clustering.
>
> + In the second class of scenarios, the results of graph clustering cannot be directly applied to the target needs, but can be assisted to enhance their effects, e.g., interest recommendation, drug discovery, knowledge graph reasoning, etc. There has been quite a bit of work on enhancing the model's ability to learn the main task by introducing additional clustering tasks, which can bring additional valid information to the main task.
>
> In summary, temporal graph clustering has rich application scenarios and can realize the effect of data enhancement and mining with richer temporal information on the basis of the original graph clustering.
>
>
> ## **Typos**
>
> + Thank you for the corrections, we will correct these errors and recheck the full article. For example, some changes are given below:
>
> > (1) Change “A sample...” to “A simple...”.
>
> > (2) Change “which we do not...” to “that we do not...”.
>
> > (3) Change “Then We conduct” to “Then, we conduct”.
>
> > (4) Change “node embeding” to “node embeddings”.
>
> > (5) Corrected the mistake with the top quote.

---

> ### Author Response · Authors · 2023-11-16
> **Response to Reviewer 8jHD [2/2]**
>
> ## **Discussion on New Datasets**
>
> Thanks to your reminder, we discuss and analyze both datasets further here. As shown in Table 4, we summarize some common public datasets, and we can see that these datasets can generally be classified into three categories.
>
> + The first category is datasets without usable labels (e.g., CollegeMsg), and in fact most of the common temporal graph datasets are of this type.
>
> + The second category is those with untrustworthy labels (e.g., Bitcoin), and we have conducted experiments on datasets in this category, and their experimental results are often lower than 5\% or even 1\%, which means that it is difficult to measure the clustering performance with such node labels.
>
> + The third category, i.e., the datasets we selected, may have some datasets with smaller sizes and some datasets with fewer timestamps, but these are already as suitable as possible for temporal graph clustering.
>
>
> However, except for our two newly developed datasets, the remaining four datasets are very limited in size, and thus the superiority of temporal graph clustering on large-scale datasets could not be verified. To address this issue, we extracted two datasets belonging to the AI domain and the CS domain from the existing OGB dataset and re-labeled these data. Combined with the experimental results on large-scale datasets in the paper, it can be found that the inclusion of these two large-scale datasets points out the limitation of depth graph clustering, i.e., the difficulty of applying the clustering method based on adjacency matrices to large-scale datasets, whereas the temporal graph method, regardless of whether it works well or poorly, can be successfully executed on these datasets.
>
> Thank you again for your professional comments and insights, which have helped us to improve the quality of our article.

---

> > ### Comment · Reviewer_8jHD · 2023-11-21
> > **Thanks for the responses!**
> >
> > Thanks for the responses. My concerns have been addressed.

---

### Author Response · Authors · 2023-11-19
**Summary**

We would like to thank all reviewers for their excellent comments, which have helped to improve the quality of our paper. We can also see that all reviewers have recognized and agreed with our paper, although there may be some confusion, we still get encouragement from the reviewers' comments.

> Due to space constraints, we have added the needed figures and tables to the last subsection (Page 21) of the appendix for review. The rest of the text will be added after a subsequent reorganization of the article layout.

Here we make a summary statement of the reviewers' confusions in an attempt to bring out the focus of the paper:

+ (1) The main contribution of this paper is not only to present a generalized framework, but also to attempt to introduce and explain the direction of temporal graph clustering in several ways (intuition, models, data, experiments, etc.).

+ (2) We focus on temporal graph clustering because it offers a new possibility of clustering based on interaction sequences (adjacency lists) compared to clustering based on adjacency matrices in discrete dynamic or static graphs. Here, we group discrete dynamic and static graphs together because they both require the entire adjacency matrix to be read for training, which can pose a serious memory overflow problem.

+ (3) Interaction sequence-based temporal graph clustering does not suffer from memory overflow because it switches to a batch processing model. By feeding the interaction records of nodes into the model in batches, it is possible to modify the batch size to avoid exceeding the memory. We therefore point out that temporal graph clustering is able to flexibly find a balance between temporal and spatial demands, which certainly provides new ideas for current graph clustering models.

+ (4) Thus, what we want to show is not that temporal graph clustering performs better compared to other methods, but that it offers new horizons. Under this premise, even if there is a gap in the performance of temporal graph clustering compared to other methods, it does not detract from the benefits of its flexibility. Therefore, the focus of our attention is not on model design, and we construct the TGC framework only to further develop experiments to support our claims.

+ (5) Finally, it is hoped that reviewers will understand that early work in a new direction is always going to be more or less inadequate because there is so much to add. And these elements need to be followed up gradually, and it is difficult to give all the discussions and results at once. Because of this, we believe that this paper, which focuses on temporal graph clustering, can still bring new perspectives and ideas to the research community of graph learning despite its slight shortcomings.

Many thanks to the reviewers for the comments, you have pointed out potential problems and helped us to further improve the paper. We also feel your recognition and approval of our work, which is encouraging, and hope that this work will be further supported by you.

---

### Meta-Review · Area_Chair_zq2i · 2023-12-08

**Metareview:**

**Summary**
This paper studies the problem of node clustering on temporal graphs. Temporal graphs have some unique properties that should be taken into account, such as dynamically changing graph topology and the restriction that earlier nodes cannot observe later nodes in chronological order. This paper achieves the integration of these properties into node clustering by combining a temporal module for time information mining and a clustering module for node clustering. While each of these components may be basic individually, their synergistic combination results in an effective strategy for the task. Empirical evaluation demonstrates the superior performance of the proposed approach.


**Strengths**
- This paper is generally well-written and easy to follow.
- This paper is well-motivated, considering that deep clustering for temporal graphs is a relatively unexplored yet relevant problem. I hope that this paper can serve as a milestone for further development in this problem setting.
- The effectiveness of the proposal has been thoroughly evaluated on experiments, showcasing superior performance compared to various existing approaches on real-world datasets.

**Weaknesses**
- There are inconsistencies in the notations, and some careful (minor) revisions are required. In Eq. (2), "$i \in N_x$" is used. According to Def. 2, $N_x = \\{(y_1, t_1), \dots, (y_l, t_l)\\}$, hence if we strictly follow this definition, "$i$" in Eq. (2) is a pair $(y_i, t_i)$. However, $\alpha((y_i, t_i), y, t)$ does not make sense, and I think this is not what the authors want to represent. The problem here is that the definition of $N_x$ in Def. 2 and notations such as $\alpha(i, y, t)$ and $\mu(i, y, t)$ are not consistent. Please correct them.

**Justification For Why Not Higher Score:**

While the proposed method is well-designed, as mentioned in the meta-review, the technical novelty of each component is not exceptionally high. Additionally, there is no theoretical analysis, and some presentation requires minor revision. Considering my opinion and the reviewers' scores, I believe "Accept (poster)" is appropriate for this paper.

**Justification For Why Not Lower Score:**

This paper is well-written and deserves to be published. During the author-reviewer discussion phase, several concerns were raised by reviewers, and the authors have successfully addressed most of them in their response. As a result, every reviewer now recommends the acceptance of the paper. Therefore, I recommend accepting the paper.

---

### Decision · Program_Chairs · 2024-01-16

Accept (poster)